# Identification of Intervention Opportunities through Assessment of the Appropriateness of Antibiotic Prescribing in Surgical Patients in a UK Hospital Using a National Audit Tool: A Single Centre Retrospective Audit

**DOI:** 10.3390/antibiotics11111575

**Published:** 2022-11-08

**Authors:** Daniel John Hearsey, Kathleen B. Bamford, Michael Hutton, Liam Wade, Henry Coates, Elizabeth Ramsay, Barbara Alberts, Neil Powell

**Affiliations:** 1Pharmacy Department, Royal Cornwall Hospital, Truro TR1 3LJ, UK; 2Medical Microbiology, Royal Cornwall Hospital, Truro TR1 3LJ, UK; 3General Surgery Department, Royal Cornwall Hospital, Truro TR1 3LJ, UK; 4Royal Cornwall Hospital, Truro TR1 3LJ, UK

**Keywords:** antibiotics, excess antibiotic use, antibiotic prescribing, antimicrobial stewardship

## Abstract

Identifying opportunities to safely reduce antibiotic prescribing is necessary for prescribers and antibiotic stewardship teams to minimise unwarranted antibiotic use. We aimed to quantify excess antibiotic use in General Surgery. We retrospectively audited the antibiotic prescribing for patients discharged from the General Surgery specialty in an acute hospital in the south-west of England for one month using an audit tool developed by Public Health England. The appropriateness of prescribing was determined for each patient at three antibiotic decision time-points: at initiation, the pre-72-h antibiotic review, and treatment duration. Two infection specialists and a general surgeon reviewed each patient. Indication and excess days of therapy (DOTs) were calculated at each decision time-point and expressed as a proportion of total DOTs. Eighty-six patients were prescribed 1162 DOTs; 192 (16.5%) excess DOTs were prescribed in 38 patients (44%), with zero excess days identified in the remaining 48 patients (56%). Seventy-five of 192 (39%) excess DOTs occurred at initiation; 55/192 (29%) after the pre-72-h antibiotic review; and 62/192 (32%) due to protracted antibiotic courses. There was concordance between the general surgeon and infection specialist for most apportioned excess DOTs. However, the surgeon apportioned fewer excess DOTs 160/1162 (13.8%). Overall IV antibiotics accounted for 53.4% of total DOTs. Seventy-two of 86 (83.7%) patients received 620 intravenous DOTs; of these, 79 (12.7%) IV DOTS were unnecessary. We have identified excess antibiotic prescribing in General surgery with comparable excess DOTs at all three time-points.

## 1. Introduction

Antimicrobial use drives antimicrobial resistance (AMR), a global problem that impacts all countries and all people, regardless of their wealth or status and is expected to result in approximately 10 million deaths each year globally by 2050 if no action is taken [1]. NHSE/I promote prudent antibiotic prescribing within NHS hospitals with implementation of multiple national quality improvement projects which aim to reduce AMR through reductions in unnecessary antimicrobial prescribing [2,3].

Antibiotic use per occupied bed-day in English hospitals varies by up to 15-fold [4]. Although there are reports across Europe for the overuse of antibiotics, the extent of overuse has not been determined; it is estimated that inappropriate prescribing accounts for 50% of all antimicrobials used in human healthcare [5]. Identifying opportunities to safely reduce antibiotic prescribing in hospitals is necessary to enable prescribers and antibiotic stewardship teams to best focus their efforts to reduce antibiotic use. However, defining the gold standard for appropriate antibiotic prescribing is challenging due to the subjective nature of evaluating quality in prescribing [6]. In 2017 the UK Government Scientific Advisory Committee on Antimicrobial Prescribing, Resistance and Healthcare Associated Infection (APRHAI) convened a workshop to define appropriate antibiotic prescribing and developed an audit tool that supports the patient-level assessment of antibiotic appropriateness in secondary care [6]. Data on antibiotic overuse in surgical specialties is limited, but study data for course optimization for complicated intra-abdominal infections demonstrates opportunity for reduction in excess antibiotic usage within general surgery [7].

The Study objectives were (1) to quantify excess antibiotic use in general surgery, measured as excess days of antibiotic therapy (DOTs) at three key time points: at initiation of antibiotics, at the pre-72-h antibiotic review, and course length, and (2) quantify excess IV antibiotic use.

## 2. Results

### 2.1. Overall

Two hundred and fifty patients were discharged from general surgery in August 2021 and were prescribed antibiotics. Of these, 39 (15.5%) patients were excluded because they were either prescribed STAT doses only or did not receive the prescribed antibiotics. Patients receiving STAT doses were excluded as these represented surgical prophylaxis and not assessed within the study. Of the remaining 211 patients, eighty-six (40.8%), with similar number of patients from each of the audit weeks, were audited.

Forty of 86 (46.5%) were female with a median age of 56 years (Interquartile Range 41–67 years). The 86 patients received 1162 DOTs with a mean of 13.5 DOTs per patient (range 1–122 days). The mean antibiotic course length per patient was 7.9 days (range 1–56 days). Of 86 infection diagnoses, 70 (86%) were surgical infections; see Table 1.

### 2.2. Excess DOTs

After the first round of review (infection specialist consensus), 223 of 1162 DOTs (19.2%) were deemed excess. After the second round of review (Consultant Surgeon (MH) and infection team representative (NP) consensus), of the 223 identified excess days; 160 were agreed upon; MH argued from a surgical perspective that 63 DOTs did not represent excess days. On further discussion NP agreed, upon reflection from an infection perspective, that 31 of these 63 DOTs could be considered not to be excess but that 32 of the 63 DOTs did still represent excess DOTs. After these discussions, 192 DOTs of 1162 (16.5%) were deemed excess days with a potential lower estimate of 160/1162 (13.8%); see Table 2.

The 192 excess DOTs were prescribed in 38 patients (44%), with zero excess days identified in the remaining 48 patients (56%). Seventy-five of 192 (39%) DOTs occurred at initiation; 55/192 (29%) at the pre-72 h antibiotic review; and 62/192 (32%) due to protracted antibiotic courses.

#### 2.2.1. Excess Antibiotics at Initiation

The 75 excess DOTs which occurred at the initiation of antibiotic therapy were prescribed in 9 of the 86 patients (10.5%). Of these, antibiotics were not indicated from the outset in 4 patients, resulting in 22 excess DOTs and in the remaining 5 patients, antibiotics were indicated at the outset, but excess DOTs were recorded due to other reasons (e.g., unnecessary duplication of antibiotic therapy) and resulted in 53 excess DOTs. There was discordance in agreement between infection specialists and surgeon for two patients resulting in 8 excess DOTs (i.e., surgeon felt antibiotic initiation was appropriate).

#### 2.2.2. Excess Antibiotics at Pre-72 h Review

Antibiotics were stopped appropriately for 26 of 86 (30.3%) patients at a pre-72-h antibiotic review. In 50 of 86 (58.1%) patients, antibiotics were appropriately continued beyond the pre-72 h review. 

The remaining 10 of 86 (11.6%) patients, inappropriate continuation of antibiotic therapy resulted in 55 excess DOTs. In 9 of the 10 cases (90%), patients’ antibiotics were continued beyond the pre-72 h review when they could have been stopped entirely, resulting in 52 excess DOTs. In the remaining patient (10%), the continuation of antibiotics was deemed appropriate but there was a missed opportunity to reduce the number of antibiotic agents being used, resulting in 3 excess DOTs.

There was discordance in agreement between infection specialists and surgeon for one patient resulting in 4 excess DOTs (i.e., surgeon felt antibiotic continuation was appropriate).

#### 2.2.3. Excess Antibiotics Due to Protracted Course Length

Antibiotics were not indicated past the standard course length for 18 patients and resulted in 61 excess DOTs. Antibiotics were indicated beyond the standard course length for one patient but resulted in 1 excess DOT due to unnecessary prolongation. 

There was discordance in agreement between infection specialists and surgeon for two patients resulting in 20 excess DOTs (i.e., surgeon felt antibiotic continuation was appropriate).

### 2.3. Excess Intravenous Antibiotic DOTs

72 of 86 (83.7%) patients received 620 intravenous DOTs; IV antibiotics accounted for 53.4% of total DOTs. Of these, 79 (12.7%) IV DOTS were excessive.

16 of 79 (20.3%) excess IV DOTs were on initiation where an antibiotic was appropriate, but the oral route would have been more appropriate (3 patients). Of the 47/79 (59.5%) at the pre-72-h antibiotic review (13 patients), 26 IV DOTs (32.9%) in 4 patients would have been more appropriate via the oral route; the remaining 16/79 (20.3%) was due to protracted course lengths where antibiotics were deemed unnecessary (1 patient).

## 3. Discussion

We have used a Public Health England (PHE) developed antibiotic appropriateness assessment tool, with infection specialists and consultant surgeon adjudication to identify that, even in an NHS trust considered a low antibiotic user overall there is still room for improvement as 16.5% of antibiotic prescribing in General Surgery was excess and therefore unnecessary. While excess DOTs occurred evenly across all three timepoints, the greatest proportion occurred at initiation of antibiotics (39%), with similar excess antibiotic DOTs occurring at the pre-72 h review (32%) and antibiotic course length (29%). 

Overuse of antibiotics for surgical patients has been reported by others [8,9]. Inappropriate antibiotic prescriptions for patients with no or little evidence of infection occurs partly due to diagnostic uncertainty; clinicians are more likely to initiate active treatment to manage this uncertainty [9,10]. A qualitative study found that surgeons perceive themselves as ‘interventionalists’ and antibiotics are prescribed for patients with no or little evidence of infection due to this need to intervene [11]. Overuse of antibiotics in the initial management of uncomplicated intra-abdominal infection occurs despite consensus agreement supporting non-antibiotic intervention once the source of infection is controlled; as well as evidence to support utilising fixed courses of antibiotics for treating complicated intra-abdominal infections once source control has been achieved [8,12]. 

Antibiotics were stopped appropriately for almost a third of patients (30.3%) at a pre-72-h antibiotic review. Despite this relatively high stop rate, we estimate this could potentially be increased to 40%, in-line with the pre-72-h antibiotic stop rate achieved by others, and similar to the potential stop rate in medical specialties within the same hospital [13,14]. Inappropriate continuation of antibiotics at the pre-72 h review accounted for almost a third of excess DOTs. Again, this is potentially driven by diagnostic uncertainty as sometimes clinicians prescribe antibiotic therapy to manage diagnostic uncertainty, leading to overprescribing of antibiotics [10]. Prescribing antibiotics in this context is reported to alleviate the risk of failure or the risk of blame, despite the potential harm from unnecessary antibiotic prescribing [10,11]. Charani et al. report a model of individualism in surgery, where surgeons driven by performance metrics, are less willing to tolerate uncertainty in antibiotic decision-making with “just in case” prescribing to protect patients of a perceived short term infection consequence [10,11]. Others have successfully increased the pre-72 h antibiotic stop rate through implementation of the Antimicrobial Review Kit (ARK) which included an online education module, a decision support tool, antibiotic hard stops at 72 h and audit and feedback cycles with associated reductions in antibiotic use and without a compromise in patient outcomes [15]. Introduction of the ARK toolkit in the study hospital could further increase the pre-72 h stop rate and is an area for further work.

Protracted course lengths account for a third of excess DOTs. Protracted course lengths put those treated with antibiotics at an increased risk of future resistant infections [16]. Antibiotic prescribing in surgical patients is driven by consideration for short-term individual patient-outcome rather than negative consequences of extended antibiotic use [17]. These consequences are often not fully appreciated because the action and the consequences are so widely separated in time that the consequences go unnoticed. The importance of prescribing antibiotics for that individual patient to prevent harm is perceived as more important than the significance of the development of antimicrobial resistance [17]. An international survey demonstrated surgeons to be aware of, and concerned by AMR, although almost half (45.6%) underrated the problem in their hospital and perceived the risk to be theoretical [18]. Despite awareness of AMR, interventions aimed to optimise antibiotic use for intra-abdominal infections is challenging and requires key stakeholder engagement to achieve sustained culture change in antimicrobial prescribing [19,20]. 

Just over half the DOTs were IV of which 12.7% were found to be unnecessary. Lack of adequate GI absorption may warrant IV therapy in patients undergoing GI surgery, however less than half of patients receiving antibiotics in surgical specialities undergo surgery and therefore the high IV therapy may not be adequately explained by poor GI absorption [9]. Defensive prescribing, with surgeons trying to avert the risk of a patient developing infections by prescribing broad-spectrum IV antibiotics may be driving the inappropriate IV antibiotic use [9]. Another explanation might be that the decision to prescribe an antibiotic sits with the consultant surgeon, but the choice of antibiotic, route and course length often delegated to the junior surgical staff who are more likely to be unaware of local guidelines or lack the knowledge or confidence to switch to oral antibiotics, resulting in unnecessary IV antibiotics or protracted course lengths [11,21]. Switching from IV to oral therapy has been shown to be safe and associated with significant benefits to clinical outcome as well reductions in adverse effects and costs [22,23].

We have identified opportunities to reduce antibiotic prescribing at all stages of the prescribing process. Empowering clinical pharmacists to challenge antibiotic prescribing in General Surgery, challenging protracted course lengths, and promoting early IV to oral switching may optimise antibiotic prescribing. Peer support and shared decision making may also increase appropriateness of antibiotic of prescribing, but optimising antibiotic prescribing is challenging [10,19]. Identification of the barriers and the enablers for surgeon and pharmacist behaviour change, and the adoption and the promotion of new prescribing practices that optimise antibiotic prescribing, requires further study. 

### Study Biases

Decisions on appropriateness were made on the information available to the prescriber at the time of prescribing using the information recorded in the medical notes and the available electronic sources: pathology, radiology, and observations. Auditors did have sight of subsequent observations and clinical test results, but these were not used to guide auditor decisions at each point of the prescribing process as they were not known to the prescriber at the time. Assessment of appropriateness is potentially subjective and so the determinations of appropriateness were made by two infection specialists until agreement sought, and then discussed with the Consultant Surgeon which adds validity to our findings. The surgeon and infection specialists largely agreed with the apportion of excess DOTs but some disagreement between infection specialists and the consultant surgeon did remain, although only in a small proportion of patients. 

This is a single centre study that reviewed only a proportion of patients in one month and therefore the results may not be representative of annual prescribing practices within general surgery nor generalisable to other hospitals. 

## 4. Materials and Methods

### 4.1. Setting

This study was conducted in a 750-bed acute secondary care hospital in the south-west of England with an electronic prescribing and medication administration system (EPMA; JAC Computer Services (WellSky), Basildon, UK) deployed in all inpatient areas. The study hospital (compared with other non-teaching hospitals in England) is a low antibiotic using hospital and ranked in the lowest quintile [24].

### 4.2. Ethics

NHS ethics approval was not required as the study did not meet the Health Research Authority definition for research or the requirements for NHS Research Ethics Committee approval. Patient data was used in accordance with local NHS hospital policy. 

### 4.3. Patient Identification

In November 2021, pharmacy data analysts used the EPMA system to identify patients who were discharged from the General Surgery specialty in August 2021 and who had received at least one dose of an antibiotic. Patients administered only a single dose or did not receive the prescribed antibiotics were excluded. Patients were grouped by week of discharge from hospital and every third patient discharged that week was included in the study; proving a number of patients that were able to audit with the available resource. Consultants rotate each week which introduces potential for variation in prescribing practices. Weekly sampling was chosen to reduce this potential bias. 

Junior doctors and antibiotic pharmacists were invited by email to participate in the data collection. Auditors were asked to read the “DEFINING AND MEASURING “APPROPRIATE” ANTIBIOTIC PRESCRIBING—A BRIEFING FOR AUDITORS” document (Appendix A) to ensure data was collected in a standardised way using the audit form (Appendix A). Auditors would complete the audit form with the relevant information from the patients’ admission record to determine the infection being managed and the antibiotic therapy that they received. After completion of the auditors first case, the audit form was emailed to NP (Consultant Antimicrobial Pharmacist) to check for completeness and correctness. Auditors were able to collect data once they were “signed off” as competent to do so by NP. Completed audit forms were emailed to NP and any discrepancies, inconsistencies, omissions, and errors were discussed with the auditor, as well as any unclear assessment on appropriateness. 

### 4.4. Data Collection

The audit form (Appendix A) facilitated collection of appropriate clinical information to determine the infection being managed and the number of antibiotic Days of Therapy (DOTs) prescribed for each patient (nearest complete 24 h). The audit form then enabled assessment of the appropriateness of prescribing at the three antibiotic decision time-points: whether antibiotic treatment was indicated from the outset, whether antibiotic treatment was indicated beyond the pre-72 h antibiotic review, whether antibiotic treatment was indicated beyond the standard treatment duration for the infection and whether IV to oral switch could have been made earlier (Appendix A for IV to oral switch criteria). 

Two infection specialists, (NP) and (KB, Medical Microbiologist), discussed each case in turn and reached consensus on excess antibiotic use. Excess antibiotic use was defined broadly as (1) Prescribing an antibiotic for a patient in the absence of (documented) evidence of signs and symptoms in keeping with a possible bacterial infection or (2) Continuing an antibiotic prescription without on-going evidence of infection (3) Continuing an antibiotic beyond the course length recommended in local or national guidelines, in the absence of a (documented) rationale” (as defined in the “Briefing for Auditors” guide (Appendix A)). Excess antibiotic use was assessed at the three time points taking into consideration the views of the junior doctor/antibiotic pharmacist auditing the case. Once consensus was reached, any cases with identified excess antibiotic use were discussed with (MH, Consultant Surgeon) to reach consensus on excess days of therapy after considering a surgeon’s viewpoint. Excess days were either agreed on or disagreed with any amendments or disagreements noted in the data collection sheet. Total antibiotic use and excess antibiotic use were measured in days of therapy (DOTs) enabling excess antibiotic use to be calculated as a percentage of total usage. 

Median and interquartile range (IQR) were used for age. Days of therapy (DOTs) and antibiotic course lengths were determined for each patient and presented as a mean and a range.

## 5. Conclusions

This study has identified several opportunities to reduce antibiotic use in General Surgery using an audit tool developed by infection specialists. The greatest opportunity to safely reduce antibiotic use was at the initiation of antibiotic stage, although there were comparable rates across all three time-points in the prescribing process.

The use of this tool has demonstrated the need and opportunity for developing further multidisciplinary effort at the time of antimicrobial prescribing decisions in surgery.

## Figures and Tables

**Table 1 antibiotics-11-01575-t001:** Summary of patient demographics.

Characteristic	Patients (*n* = 86)
Sex	
Male	46 (53.5%)
Female	40 (46.5%)
Age (years)	
Median	56
Inter Quartile Range (IQR)	26
Audit week	
Week 1 (2nd August–8th August)	23 (26.7%)
Week 2 (9th August–15th August)	25 (29.1%)
Week 3 (16th August–22nd August)	18 (20.9%)
Week 4 & 5 (23rd August–31st August)	20 (23.3%)
Infection diagnosis	81
**Surgical**	**70**
*Abscess*	
Complex Pelvic	2
Gluteal	1
Hepatic	3
Anorectal	3
Pilonidal	2
Thigh	2
Appendicitis	9
Bowel Obstruction	6
Cholangitis	3
Cholecystitis	12
Colitis	3
Contaminated wound	2
Diverticulitis	4
Fistula	2
Gastroenteritis	2
Haemorrhoids	1
Infected Sebaceous Cyst	1
Pancreatitis	1
Perforated bowel	3
*Surgical Procedure*	
Cholecystectomy	1
Extra-levator abdominal perineal resection	1
Hartmanns	1
Polyp removal	2
Extended Surgical Prophylaxis	3
**Non-surgical**	**11**
Clostridioides difficile associated diarrhoea	1
Hospital-acquired pneumonia	3
Pyelonephritis	2
Sepsis (undifferentiated)	2
Urinary tract infections	3
**No final infection diagnosis**	5

**Table 2 antibiotics-11-01575-t002:** Excess antibiotic days of therapy at each antibiotic decision time-points at each stage of specialist consensus.

Source		Excess DOTs				Percentage Excess DOTs of Total DOTs
	Total DOTs	At Antibiotic Initiation	At the Pre-72 h Antibiotic Review	Antibiotic Course Length	Total	
Infection Specialist consensus	1162	82	68	73	223	19.9%
Consultant Surgeon and infection specialist consensus	1162	75	55	62	192	16.5%
Consultant surgeon determined excess days (with some disagreement between infection specialists and surgeon)	1162	67	51	42	160	13.8%

## Data Availability

Data available on reasonable request from corresponding author.

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
