# Peer review of "Identification of Intervention Opportunities through Assessment of the Appropriateness of Antibiotic Prescribing in Surgical Patients in a UK Hospital Using a National Audit Tool: A Single Centre Retrospective Audit"

_antibiotics, 2022, doi:10.3390/antibiotics11111575_

Round 1
Reviewer 1 Report
Its an interesting paper to read but I have some minor suggestions.
Why small sample size, maybe it hinder the findings.
Improve study rationale in the introduction section and discuss background using previous studies.
Cleary write objective of the study.
Its better to have sub-headings in the method section because I think it easier for the public to read.
thanks
Reviewer 2 Report
Can be seen in attached file

Reviewer 3 Report
This manuscript is a retrospective study on the prescription of antibiotics and the excessive use of antibiotics through excess days of antibiotic therapy (DOTs) through a standardized audit form.
The most significant deficiency is that not all the methods used to obtain the obtained results are highlighted, and this leads to the fact that it is difficult to understand what the article is about (difficult to read and understand). The article must be modified for readability.
The randomization criteria of the Two hundred and fifty patients are not mentioned; there are different methods to avoid bias in sampling. You must declare which one was used.
Methodology not mentioned:
It is not mentioned how the Audit week was carried out (week 1, week 2, week 3 and week 4)
It is not mentioned that the interquartile range (IQR) will be used, nor is the acronym's meaning mentioned.
The use of ranges is not mentioned.
There is no mention of the consensus process of (infection specialist consensus), and (Consultant Surgeon 71 (MH) and infection team representative (NP) consensus) and how biases are avoided. (Because each doctor has a different point of view)
It is not mentioned in the methodology that there will be 3 different times (also, it is not homologated in table 2; it appears as timepoint A, B and C and in the text as; initiation, pre-72-98 hour and treatment duration)
It does not delve into how the DOTs were obtained (it is essential since it is the primary indicator of the study)
The objective “IV to oral switch was also assessed” is not mentioned as it will be done.
The structure of having two objectives “1) excess antibiotic use, measured as excess days of antibiotic 53 therapy (DOTs); 2) the potential for IV to oral switch was also assessed, it is confusing.
The tables do not explain the acronyms, for example, IQR
In Table 1, Non-surgical and Surgical are in bold; please add this in No final infection diagnosis
Add study biases
Missing to mention the acronym "PHE"
Why are patients who were prescribed the STAT drug discarded?
What is the role of the DDDs in the development of the study?
In addition, material and methods are not where that information should be described.
References 21 and 22 do not appear in the manuscript.
Round 2
Reviewer 3 Report
No more further comments